# Towards Trustworthy and Identifiable Virtual Face Generation

Chunyang Li [* 1]  Haoyue Wang [* 1]  Zhenxing Qian [1]  Sheng Li [1]  Xinpeng Zhang [1]  Jian Liu [2]  Yixuan Pei [2]  Weiqiang Wang [2]

## Abstract

Identifiable virtual face (IVF) generation aims to transform a user's original face into a virtual face for high utility privacy protection. The IVF is visually and statistically different from the original face, which can still be used for recognizing the user's identity. Despite this advantage, these schemes are unable to verify the trustworthiness of the IVF, the quality and controllability of which is often limited. To address these issues, we propose TIVDiff, a diffusion-based framework for trustworthy and identifiable virtual face generation. TIVDiff learns a virtual identity (VID) space via Virtual Identity Projection (VIP) and synthesizes high-quality virtual faces conditioned on VID and 3D facial geometry for pose and expression preservation. To enable the trustworthiness of IVF, we further propose an Identity-Guarded Generative Watermarking (IGGW) scheme to bind the diffusion initial noise with VID through a reversible mapping mechanism. This enables the embedding of an imperceptible cue into IVF for legitimacy verification. Experiments demonstrate the advantage of our TIVDiff over the state-of-the-art IVF generation schemes in terms of image quality, identifiability and trustworthiness.

## 1. Introduction

With the rapid advancement in digital technologies and smart devices, face-based applications are now widely deployed in various fields, e.g., payment, identity verification, and access control. However, accompanying the convenience is the growing concern over the large-scale, non-consensual collection and circulation of face images which are linked to person's identity and irrevocable. Once leaked, the identity information may be leveraged by adversaries for malicious purposes, such as identity impersonation and fraudulent authentication.

To protect face privacy, early works apply obfuscation-based techniques, such as mosaicing, blurring, and pixelation (Newton et al., 2005), to anonymize face images. However, these operations often induce severe visual degradation, which hinders subsequent applications. With the rise of generative models (Goodfellow et al., 2020; Rombach et al., 2022), researchers have explored synthesizing surrogate faces to anonymize the original face (Maximov et al., 2020; Yang et al., 2024a; Kung et al., 2025), but the original identity is typically irreversibly discarded, preventing identification when needed. To better balance privacy and utility, subsequent studies aim to preserve identity while altering appearance, e.g., via attribute editing to generate visually different yet still identifiable faces (Li et al., 2021), or by leveraging data hiding to embed identity cues into protected faces (Yuan et al., 2022a). Another strategy is to encrypt a face image into a real look-alike anonymous face and later recover the original face with a secret key for authentication (Zhang et al., 2023b; Yuan et al., 2025; Zhu et al., 2025). Nevertheless, these methods still rely on physical identity (PID) for face recognition, which has risks of privacy leakage.

Recently, identifiable virtual faces (IVF) associated with a virtual identity (VID) have emerged as a promising substitute for original faces in downstream face recognition applications, offering strong privacy protection by decoupling VID from PID. IVFG (Yuan et al., 2022b) maps each PID to a VID, which is then fed into a generative adversarial network (GAN) for IVF generation. KFAAR (Wang et al., 2025) further proposes an identity attribution module to recover the PID from the VID for traceability. CanFG (Wang et al., 2024) and FaceAnonyMixer (Alam et al., 2025) preserve identity-irrelevant attributes to support downstream tasks such as pose analysis. These methods achieve satisfactory identifiability and high privacy protection level. However, they lack a mechanism to verify the legitimacy of the IVF. The administrator is unable to verify whether it is a fake face (generated by an attacker) or a legitimate virtual face (generated by a user). This severely limits the application of IVF for real-world deployment.

*Equal contribution [1]Fudan University [2]Ant Group. Correspondence to: Sheng Li <lisheng@fudan.edu.cn>.

*Proceedings of the 43rd International Conference on Machine Learning*, Seoul, South Korea. PMLR 306, 2026. Copyright 2026 by the author(s).

For IVF to be reliably and seamlessly integrated into the existing face recognition systems, we should study how we could ensure the trustworthiness of the IVF to prevent the system from the fake face attacks. It also requires the IVF to have the same expression and pose as the original face to meet the needs of challenge-response face recognition systems. To this end, we propose to embed an intrinsic, privacy-preserving verification cue during IVF generation. This enables the system to verify the legitimacy of IVF without recovering the PID. Instead of GAN-based IVF generation which has been studied in literature, we explore diffusion-based IVF generation to improve the quality of the IVF with the face expression and pose well maintained.

Concretely, we propose a Trustworthy Identifiable Virtual Face Diffusion (TIVDiff) framework in this paper. In TIVDiff, we first train an effective Virtual Identity Projection (VIP) module to map the PID to diverse VIDs while preserving cross-instance consistency via a specific key. For high-quality and controllable IVF generation, we then incorporate Arc2Face (Papantoniou et al., 2024) for the design of a Pose-Aware Identity-Conditioned Diffusion (PoseID-Diff) module which takes the VID and the 3D facial geometry of the original face as conditions to generate an IVF. Meanwhile, we propose an Identity-Guarded Generative Watermarking (IGGW) scheme to establish a reversible mapping between the VID and the diffusion initial noise. This serves as an intrinsic watermark for legitimacy verification of the IVF, and is robust against common post-processing and watermark forgery attacks. Experiments show that TIVDiff achieves superior image quality and identifiability over the state-of-the-art (SOTA) IVF methods. We also provide a security analysis of potential attacks to substantiate the robustness and trustworthiness of the proposed method. The main contributions are summarized below.

- We introduce a trustworthy paradigm for IVF generation by embedding intrinsic, privacy-preserving verification cues, making legitimacy verifiable without recovering the PID.

- We explore the potential of diffusion models for IVF generation and propose TIVDiff to generate high-quality, controllable, and identifiable virtual faces.

- We design a new loss function to train an effective Virtual Identity Projection (VIP) module to map each PID to realistic, diverse, and selectable VIDs with cross-instance consistency.

- We propose IGGW, a watermarking scheme to reversibly couple VID with diffusion initial noise for legitimacy verification of the IVF, which is robust against post-processing and watermark forgery attacks.

## 2. Related Work

### 2.1. Face Anonymization

Early face anonymization methods apply image obfuscation operations (Newton et al., 2005; Gross et al., 2006), while easy to deploy, they often cause substantial utility loss and remain vulnerable to re-identification attacks (McPherson et al., 2016).

To better preserve visual quality, a line of work leverages generative models to synthesize high-fidelity surrogate faces for anonymization. CIAGAN (Maximov et al., 2020) explicitly replaces identity while retaining task-relevant cues such as pose, expression, and background context. Deep-Blur (Li & Choi, 2021) suppresses identity features via a pretrained generator, producing visually natural outputs. These methods can produce faces that appear anonymous and are difficult to re-identify by face recognizers. Another line of work aims to preserve visual appearance while achieving anonymity against face recognizers, typically by leveraging adversarial perturbations. TIP-IM (Yang et al., 2021) overlays targeted identity masks optimized via iterative procedures. Follow-up methods (Yin et al., 2021; An et al., 2023) further improve stealthiness by embedding perturbations into naturalistic edits, enhancing visual realism and black-box transferability. Despite their effectiveness, both lines largely discard the PID, making the anonymized faces difficult to use in downstream face-related applications that require identity continuity.

To balance privacy and utility, some works (Yuan et al., 2022a; Yang et al., 2025) preserve PID compatibility by modifying identity-irrelevant attributes. Other works (Pan et al., 2021; Zhang et al., 2023b; Zhu et al., 2025) explore reversible anonymization, where the anonymized face can be restored to the original one for identity verification when needed. However, these approaches still rely on PID for recognition, which inherently carries privacy risks.

Recently, some works (Yuan et al., 2022b; Wang et al., 2024; Alam et al., 2025) attempt to replace original faces with IVF for anonymization. The original face is intended to be irrecoverable, and a VID is used in place of PID for downstream applications. While prior studies demonstrate the promise of this paradigm, existing GAN-based methods often struggle to achieve both high fidelity and flexible controllability. Moreover, the legitimacy of IVF is typically not verifiable, leaving VID-based applications vulnerable to unauthorized faces and limiting real-world deployment.

### 2.2. Generative Watermarking

Generative watermarking embeds an imperceptible signature during image synthesis, enabling copyright attribution and ownership verification of the generated content. Early efforts primarily follow a training-based paradigm (Fernan-

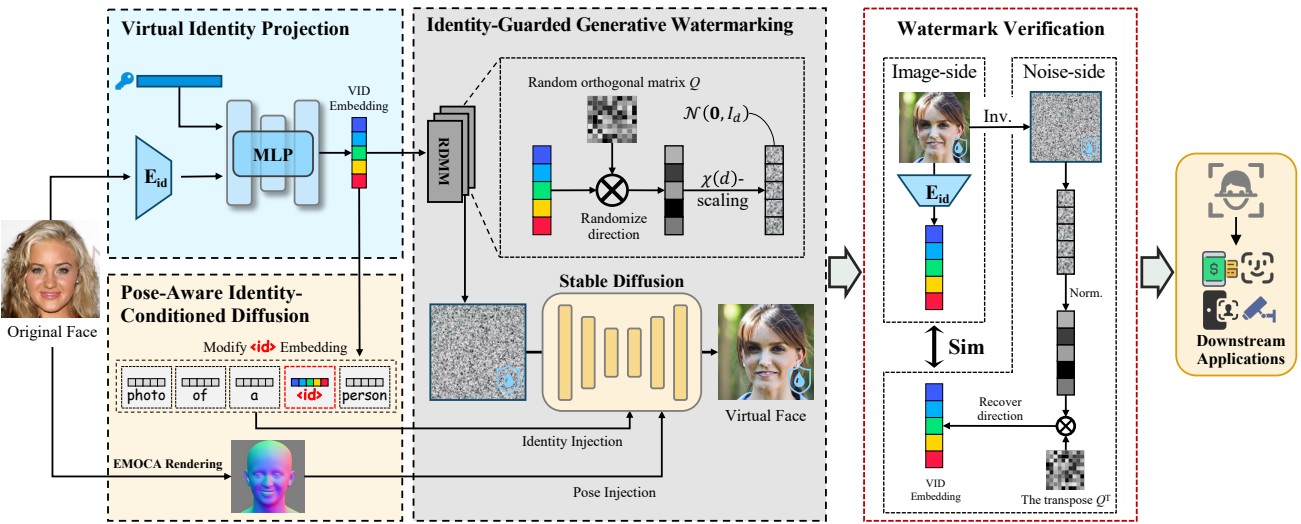

*Figure 1.* **Overview of TIVDiff.** TIVDiff comprises three modules: (1) **VIP**, which maps a PID to a selectable VID embedding; (2) **PoseID-Diff**, which injects the VID embedding and a facial normal map into Stable Diffusion to generate virtual faces while preserving pose and expression; and (3) **IGGW**, which binds the VID embedding to the diffusion initial noise via a Reversible Distribution-Matched Mapping (RDMM) for legitimacy verification.

dez et al., 2023; Zhang et al., 2024; Sander et al., 2025), fine-tuning model components or attaching lightweight modules so generations become watermark-carrying by design. Subsequent works explore a training-free paradigm (Wen et al., 2023; Ci et al., 2024; Yang et al., 2024b; Hu et al., 2024; Gunn et al., 2025; Arabi et al., 2025), and they inject a fixed pattern into diffusion initial noise, thereby embedding the watermark without extensive model retraining.

Despite this progress, generative watermarks remain fragile under watermark forgery attacks (Müller et al., 2025), which allow adversaries to transplant a valid watermark onto unrelated images. In our framework, watermarking serves as a security primitive for legitimacy verification, instead of provenance, copyright attribution, or forensics. This setting is inherently adversarial, because a static pattern can be copied and reused across samples to bypass verification.

## 3. Method

### 3.1. Overview

The overall framework of our method is illustrated in Fig. 1. Given an original face image, we first extract its PID embedding and obtain a VID embedding via Virtual Identity Projection (VIP). The VID embedding, together with the 3D facial geometry of the original face, is then injected into Pose-Aware Identity-Conditioned Diffusion (PoseID-Diff) to guide the diffusion process to synthesize an identifiable virtual face (IVF) that follows the pose and expression of the original face, supporting challenge-response requirements. Meanwhile, Identity-Guarded Generative Watermarking (IGGW) is introduced to embed an intrinsic verifi-

cation cue by mapping the VID embedding to the diffusion initial noise. During verification, we invert a queried image to recover its initial noise and apply the inverse mapping to obtain a noise-implied VID embedding. Verification succeeds if it matches the VID embedding extracted from the queried image.

### 3.2. Virtual Identity Projection

To enable effective privacy protection while preserving usability, we aim to generate an IVF whose identity is distinct from the original face. Motivated by the work in (Yuan et al., 2022b), we design here a Virtual Identity Projection (VIP) module, which is able to generate virtual identities for different scenarios without identity linkage. Concretely, the VIP maps the PID to a selectable VID embedding, where a user-specific key determines the VID selection and encourages strong separation between VIDs, enhancing privacy by preventing cross-VID linkage. We provide a detailed explanation of the VIP as follows.

**Face Embedding Extraction.** As illustrated in Fig. 1, given an original face image $I_i \in \mathbb{R}^{H \times W \times C}$, where $H$, $W$, $C$, and $i$ denote the height, width, number of channels, and sample index, respectively, we extract its PID embedding $\phi_i \in \mathbb{R}^d$ using an existing face image encoder $E_{id}$, which is formulated as

$$\phi_i = E_{id}(I_i). \tag{1}$$

**Identity Projection.** To generate diverse virtual faces from the same PID embedding $\phi_i$, we introduce a key-controlled mechanism that steers the projector to map $\phi_i$ to different VID embeddings. The projector is implemented as a

Multi-Layer Perceptron (MLP) $P(\cdot)$, which produces a VID embedding conditioned on the selected key $\mathbf{k}_m$,

$$\gamma_i^m = P(\phi_i, \mathbf{k}_m), \tag{2}$$

where $m \in [1, S]$ indexes different keys. By choosing different $\mathbf{k}_m$, VIP can generate $S$ distinct VID embeddings from the same PID embedding. Meanwhile, for a fixed $\mathbf{k}_m$, the projected VID embeddings remain consistent across different samples of the same PID, enabling a stable VID assignment.

**Loss Functions for VIP Training.** We optimize VIP with five loss terms, each corresponding to a desired property: (i) anonymization, (ii) diversity, (iii) intra-identity consistency, (iv) inter-identity separability, and (v) manifold regularization. We define each term below.

*Anonymization Loss.* To suppress biometric identity leakage, we push the VID embedding away from the corresponding PID embedding

$$L_{\text{anony}} = \ell_{\cos}\big(\phi_i, \gamma_i^m; -0.1\big). \tag{3}$$

where $\ell_{\cos}(\mathbf{a}, \mathbf{b}; y)$ is a cosine-based loss that encourages the cosine similarity between $\mathbf{a}$ and $\mathbf{b}$ to approach the target $y$. Here we set $y$ to a small negative value to enhance anonymization.

*Diversity Loss.* To achieve diversity across keys for the same PID, we enforce the VID embeddings induced by two different keys to be dissimilar

$$L_{\text{div}} = \ell_{\cos}\big(\gamma_i^{m_1}, \gamma_i^{m_2}; 0\big), \quad m_1 \neq m_2. \tag{4}$$

*Intra-identity Loss.* To keep a stable VID under a fixed key, we pull together projections of the same PID embedding

$$L_{\text{intra}} = \ell_{\cos}\big(\gamma_p^m, \gamma_q^m; 1\big). \tag{5}$$

*Inter-identity Loss.* To preserve inter-person discriminability under the same key, we push apart VID embeddings of different PIDs

$$L_{\text{inter}} = \ell_{\cos}\big(\gamma_p^m, \gamma_r^m; 0\big). \tag{6}$$

Here $p$ and $q$ denote two samples of the same PID, while $r$ denotes a sample from a different PID.

*Distribution Regularization Loss.* To keep VID embeddings on the face embedding manifold, instead of using a simple L2 norm as in (Yuan et al., 2022b), we design a distribution regularization loss here to make the VIDs more realistic. In particular, we fit a PCA model on a public face dataset and define a reference Gaussian distribution in the PCA space. Let $(\mu_r, \Sigma_r)$ be the mean/covariance of real embeddings and $(\mu_v, \Sigma_v)$ be batch statistics of projected embeddings in the same PCA space:

$$L_{\text{dis}} = D_{\text{KL}}(\mathcal{N}(\mu_v, \Sigma_v) \,\|\, \mathcal{N}(\mu_r, \Sigma_r)). \tag{7}$$

The final training objective is

$$\begin{aligned} L &= \lambda_{\text{anony}} L_{\text{anony}} + \lambda_{\text{div}} L_{\text{div}} + \lambda_{\text{intra}} L_{\text{intra}} \\ &\quad + \lambda_{\text{inter}} L_{\text{inter}} + \lambda_{\text{dis}} L_{\text{dis}}. \end{aligned} \tag{8}$$

where $\lambda_{\text{anony}}, \lambda_{\text{div}}, \lambda_{\text{intra}}, \lambda_{\text{inter}}$, and $\lambda_{\text{dis}}$ are the weighting coefficients for the corresponding loss terms.

### 3.3. Pose-Aware Identity-Conditioned Diffusion

Unlike traditional methods for generating virtual faces based on GAN, we incorporate Arc2Face (Papantoniou et al., 2024) for Pose-Aware Identity-Conditioned Diffusion (PoseID-Diff) to generate high-quality virtual faces while preserving the pose and expression of the original face to meet challenge-response requirements. Given the VID embedding $\gamma_i^m$, we control the generation of virtual faces with the associated VID, while introducing the pose condition that explicitly controls the facial attributes of the virtual face. The detailed explanation is as follows.

**Identity Injection.** We use a fixed prompt template, "photo of a <id> person". After tokenization, we replace the <id> token embedding with the VID embedding $\gamma_i^m$. The modified prompt is then fed into the CLIP text encoder (Radford et al., 2021) to obtain the identity condition. Then, the condition is injected into the diffusion model.

**Pose Injection.** We use EMOCA (Daněček et al., 2022) to regress the FLAME (Li et al., 2017) parameters of the original face image, from which we render facial normal maps that serve as identity-agnostic pose conditions while remaining sensitive to pose and expression variations. This normal map is injected via a ControlNet (Zhang et al., 2023a) branch, which guides virtual face generation to preserve original head pose and facial expression.

By injecting these two conditions, PoseID-Diff synthesizes identifiable virtual faces (IVF) that preserve the original pose and expression, while ensuring that the generated faces align with the specified VID.

### 3.4. Identity-Guarded Generative Watermarking

PoseID-Diff generates high-quality IVFs, but their legitimacy cannot be verified. For example, equally high-quality face images can also be generated by unauthorized generative models or deepfakes, which are often unacceptable for real-world applications. To this end, we propose an Identity-Guarded Generative Watermarking (IGGW) scheme. In IGGW, we bind the diffusion initial noise with the VID embedding $\gamma$ through a newly designed Reversible Distribution-Matched Mapping (RDMM), ensuring that the generated IVF based on this initial noise carries an intrinsic and verifiable legitimacy marker. Thus, downstream applications can provide services only for legitimate IVFs, preventing the impact of unauthorized synthetic faces and making IVF

a reliable solution for privacy protection. In the following, we provide a detailed explanation.

**Reversible Distribution-Matched Mapping (RDMM).** We introduce RDMM as a modular transform that establishes a reversible mapping between a VID embedding and a Gaussian embedding (i.e., Gaussian-distributed noise). Let $\gamma \in \mathbb{R}^d$ be the unit-normalized VID embedding produced by VIP, with $\|\gamma\|_2 = 1$ and $d=512$. We randomly initialize an orthogonal matrix $Q \in \mathbb{R}^{d \times d}$ satisfying $Q^\top Q = I$. RDMM maps $\gamma$ to a Gaussian embedding by

$$\mathbf{z}_0 = r \cdot (Q\gamma), \qquad r \sim \chi(d), \qquad (9)$$

and supports inverse direction recovery via

$$\hat{\gamma} = Q^\top \frac{\mathbf{z}_0}{\|\mathbf{z}_0\|_2}. \qquad (10)$$

This construction preserves the target distribution. A standard Gaussian embedding $\mathbf{z} \sim \mathcal{N}(\mathbf{0}, I_d)$ admits the polar decomposition $\mathbf{z} = r\mathbf{u}$, where $\mathbf{u}$ is uniformly distributed in the unit sphere and $r \sim \chi(d)$ is an independent radius. We apply a random orthogonal matrix $Q$ to pseudo-randomize the direction of $\gamma$ and sample a Gaussian-consistent radius $r$, yielding $\tilde{\mathbf{z}} = r \cdot (Q\gamma)$. Since $Q$ preserves isotropy and the radius matches that of a standard Gaussian, $\tilde{\mathbf{z}}$ approximately matches $\mathcal{N}(\mathbf{0}, I_d)$, while it can be reversibly verified using $Q^\top$.

**Watermark Embedding.** For each generated image, we apply RDMM independently per chunk:

$$\mathbf{z}_0^c = r_c \cdot (Q^c \gamma), \qquad r_c \sim \chi(d), \quad c = 1, \ldots, B, \quad (11)$$

where $\mathbf{z}_0^c \in \mathbb{R}^d$, $d=512$, and $Q^c \in \mathbb{R}^{d \times d}$ is an orthogonal matrix associated with chunk $c$. We use different $Q^c$ across chunks to avoid repeated initialization patterns, which may otherwise introduce structured image artifacts. Here $B$ is a fixed constant determined by the initial noise size and the chunk length. Since the initial noise has shape $4 \times 64 \times 64$, the maximum capacity is $B \leq \lfloor \frac{4 \cdot 64 \cdot 64}{512} \rfloor = 32$. We then pack the $B$ chunks into the full initial noise by concatenation followed by reshaping:

$$\mathbf{z}_0 = \text{reshape}\Big(\text{concat}\big(\mathbf{z}_0^1, \ldots, \mathbf{z}_0^B\big)\Big) \in \mathbb{R}^{4 \times 64 \times 64}. \quad (12)$$

Under the fixed reshape ordering, each 512-D chunk occupies a contiguous subset of initial noise, so the watermark signal is spread across the entire image, while aggregating evidence over chunks improves robustness to common post-processing.

**Watermark Verification.** Given a queried image $\hat{I}$, we employ an inversion operation to estimate its initial noise as $\hat{\mathbf{z}}_0$ and then reshape it as

$$\hat{\mathbf{Z}}_0 = \text{reshape}(\hat{\mathbf{z}}_0) \in \mathbb{R}^{4 \times 64 \times 64}. \quad (13)$$

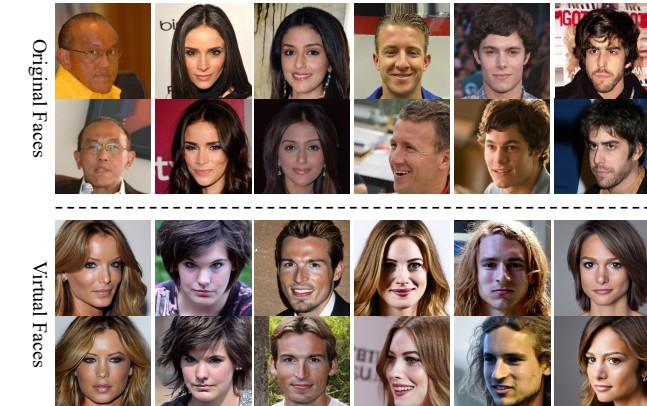

*Figure 2.* Examples of our virtual face images generated from the CelebA test set. The first row shows the original face images, and the corresponding virtual faces are presented in the second row.

We then split $\hat{\mathbf{Z}}_0$ to get $B$ chunks $\{\hat{\mathbf{z}}_0^c\}_{c=1}^B$, each chunk is a 512-D embedding. We also extract an image-side identity embedding $\gamma_{\text{img}} = E_{id}(\hat{I})$.

We recover per-chunk identity directions by applying $Q^\top$

$$\hat{\gamma}^c = (Q^c)^\top \frac{\hat{\mathbf{z}}_0^c}{\|\hat{\mathbf{z}}_0^c\|_2}, \qquad c = 1, \ldots, B. \quad (14)$$

We compare each recovered direction to $\gamma_{\text{img}}$ using cosine similarity and aggregate chunk evidence:

$$s(\hat{I}) = \text{Agg}_{c \in [B]}\big(\cos(\hat{\gamma}^c, \gamma_{\text{img}})\big). \quad (15)$$

Finally, we accept the image as legitimate if $s(\hat{I})$ exceeds a preset threshold.

It is worth noting that IGGW verification is not based solely on the presence of the watermark, but on the binding consistency between the noise-side watermark evidence and the image-side identity. Therefore, for watermark forgery attacks, such as inverting a legitimate face image to its watermarked noise and regenerating, or performing careful face swapping to retain the watermark, IGGW is capable of resisting such attacks.

## 4. Experiments

### 4.1. Experimental Setup

**Datasets.** We train our model on CelebA (Liu et al., 2015) and evaluate it on both CelebA and FFHQ (Karras et al., 2019). We perform an identity-disjoint split on CelebA, 90% of identities are used for training and the remaining 10% for in-domain testing, with no subject overlap between the training and test sets. To evaluate cross-dataset generalization, we additionally sample a subset from FFHQ as an out-of-domain test set.

**Implementation details.** We use ArcFace (Deng et al.,

*Table 1.* Virtual face identifiability of different methods.

| Methods | ArcFace | | AdaFace | | Average | |
|---|---|---|---|---|---|---|
| | EER ↓ | AUC ↑ | EER ↓ | AUC ↑ | EER ↓ | AUC ↑ |
| Original | 0.052 | 0.976 | 0.055 | 0.978 | 0.054 | 0.977 |
| IVFG | 0.101 | 0.962 | 0.102 | 0.960 | 0.102 | 0.961 |
| DP2 | 0.329 | 0.732 | 0.301 | 0.764 | 0.315 | 0.748 |
| RiDDLE | 0.111 | 0.950 | 0.109 | 0.952 | 0.110 | 0.951 |
| KFAAR | 0.262 | 0.816 | 0.265 | 0.813 | 0.263 | 0.815 |
| G$^2$Face | 0.411 | 0.629 | 0.434 | 0.597 | 0.422 | 0.613 |
| FAS | 0.381 | 0.666 | 0.365 | 0.687 | 0.373 | 0.677 |
| **Ours** | **0.036** | **0.988** | **0.045** | **0.986** | **0.041** | **0.987** |

*Table 2.* Anonymization results. Protection rate (PR) using EER threshold and mean cosine similarity (mCos).

| | Methods | ArcFace | | AdaFace | | Average | |
|---|---|---|---|---|---|---|---|
| | | PR ↑ | mCos ↓ | PR ↑ | mCos ↓ | PR ↑ | mCos ↓ |
| CelebA | IVFG | 0.954 | 0.005 | 0.991 | -0.003 | 0.973 | 0.001 |
| | DP2 | 0.680 | 0.069 | 0.387 | 0.171 | 0.534 | 0.120 |
| | RiDDLE | 0.905 | 0.016 | 0.863 | 0.065 | 0.884 | 0.041 |
| | KFAAR | 0.946 | 0.011 | 0.950 | 0.038 | 0.948 | 0.024 |
| | G$^2$Face | 0.762 | 0.049 | 0.871 | 0.051 | 0.816 | 0.050 |
| | FAS | 0.595 | 0.088 | 0.420 | 0.167 | 0.508 | 0.128 |
| | **Ours** | **0.979** | **-0.024** | **0.981** | **0.002** | **0.980** | **-0.011** |
| FFHQ | IVFG | 0.964 | -0.005 | 0.988 | 0.001 | 0.976 | -0.002 |
| | DP2 | 0.559 | 0.088 | 0.503 | 0.148 | 0.531 | 0.118 |
| | RiDDLE | 0.854 | 0.036 | 0.841 | 0.070 | 0.848 | 0.053 |
| | KFAAR | 0.954 | 0.006 | 0.982 | 0.016 | 0.968 | 0.011 |
| | G$^2$Face | 0.621 | 0.079 | 0.878 | 0.046 | 0.749 | 0.062 |
| | FAS | 0.318 | 0.154 | 0.337 | 0.202 | 0.328 | 0.178 |
| | **Ours** | **0.986** | **0.001** | **0.990** | **-0.008** | **0.988** | **-0.004** |

*Table 3.* Stronger privacy attack results on FFHQ. Lower is better except AUC, where values closer to chance indicate weaker leakage.

| Method | AUC | L-R1↓ | Cls.↓ | Ret.↓ | R-R1↓ |
|---|---|---|---|---|---|
| IVFG | 0.6052 | 0.0323 | 0.0110 | 0.0018 | 0.0270 |
| RiDDLE | 0.8245 | 0.7455 | 0.6170 | 0.0245 | 0.8620 |
| KFAAR | 0.9806 | 0.8512 | 0.7930 | 0.0010 | 0.7600 |
| G$^2$Face | 0.8120 | 0.3845 | 0.4600 | 0.0163 | 0.9620 |
| **Ours** | **0.4769** | **0.0027** | **0.0010** | **0.0003** | **0.0220** |

AUC and EER of different methods are reported in Table 1, with the AUC and EER of the original faces also included for reference. It can be observed that RiDDLE, KFAAR, IVFG, and our method achieve relatively high identifiability, with our method achieving the best performance, reducing EER by approximately 6% compared to the second-best (i.e., IVFG). Notably, our method even surpasses the identifiability level measured on the original face images, suggesting that the proposed virtual identity projection produces compact and stable identity representations.

Fig. 2 shows a few examples of our method, as well as the corresponding original ones from the CelebA test set. This qualitative consistency aligns with the low EER and high AUC, indicating the excellent identifiability of our method.

**Anonymization.** We use protection rate (PR) to evaluate the anonymization effectiveness of different methods, where the PR is measured as the mismatch rate between original face images and corresponding virtual face images. Specifically, an original face image and the corresponding virtual face image are mismatched if the cosine similarity is less than the EER threshold of the face recognizer. We additionally provide the mean cosine similarity (mCos) as a threshold-independent indicator of anonymization effectiveness. As shown in Table 2, IVFG, KFAAR, and our method achieve relatively strong PR, while our method achieves the best performance on both CelebA and FFHQ.

We further evaluate whether the generated virtual faces leak recoverable PID information under three attacks, including 1) cross-key linkage, which assesses whether virtual faces generated from the same PID with different keys remain linkable. We report AUC for pairwise verification and rank-1 accuracy (i.e. L-R1) retrieval of the cross-key virtual face with the same PID. 2) PID inference, which evaluates whether an attacker can infer the original PID from a virtual face. Top-1 classification accuracy (i.e., Cls.) and rank-1 retrieval accuracy (i.e., Ret.) are reported against the original PID feature gallery. And 3) PID reconstruction, which measures whether the original PID can be recovered from reconstructed PID features, is reported by rank-1 accuracy (i.e., R-R1). All attacks are conducted on FFHQ using AdaFace as the attacker's identity encoder. As shown in Table 3, our method substantially reduces cross-key linkability

2019) as the identity encoder $E_{id}$ for all identity-space measurements. PoseID-Diff is built upon a pretrained Stable Diffusion v1.5, with weights initialized from Arc2Face (Papantoniou et al., 2024). We set the number of keys $S$=16, and the weights of loss $\lambda_{\text{anony}}$=0.1, $\lambda_{\text{div}}$=1, $\lambda_{\text{intra}}$=1, $\lambda_{\text{inter}}$=1, and $\lambda_{\text{dis}}$=2.

**Baselines.** For anonymization, we include 6 representative face anonymization methods, including IVFG (Yuan et al., 2022b), DP2 (Hukkelås & Lindseth, 2023), RiDDLE (Li et al., 2023), KFAAR (Wang et al., 2025), G$^2$Face (Yang et al., 2024a), and FAS (Kung et al., 2025). For watermarking, we compare against 4 generative watermarking methods, including Tree-Ring (Wen et al., 2023), Gaussian Shading (Yang et al., 2024b), WIND (Arabi et al., 2025), and PRC (Gunn et al., 2025).

### 4.2. Main Results

**Virtual face identifiability.** We generate multiple virtual face images of the same PID using different methods and evaluate their identifiability with two widely adopted face recognizers, ArcFace and AdaFace (Kim et al., 2022). The

*Table 4.* Diversity results. Diversity (Div.) using EER threshold and mean cosine similarity (mCos).

| | Methods | ArcFace | | AdaFace | | Average | |
|---|---|---|---|---|---|---|---|
| | | Div. ↑ | mCos ↓ | Div. ↑ | mCos ↓ | Div. ↑ | mCos ↓ |
| CelebA | IVFG | 0.287 | 0.170 | 0.655 | 0.117 | 0.471 | 0.144 |
| | RiDDLE | 0.200 | 0.218 | 0.309 | 0.230 | 0.255 | 0.224 |
| | KFAAR | 0.031 | 0.557 | 0.069 | 0.567 | 0.060 | 0.562 |
| | G$^2$Face | 0.090 | 0.380 | 0.186 | 0.346 | 0.138 | 0.363 |
| | **Ours** | **0.881** | **0.025** | **0.928** | **0.037** | **0.905** | **0.031** |
| FFHQ | IVFG | 0.259 | 0.179 | 0.605 | 0.132 | 0.432 | 0.156 |
| | RiDDLE | 0.068 | 0.357 | 0.173 | 0.374 | 0.121 | 0.366 |
| | KFAAR | 0.020 | 0.575 | 0.058 | 0.585 | 0.039 | 0.580 |
| | G$^2$Face | 0.146 | 0.354 | 0.253 | 0.312 | 0.199 | 0.333 |
| | **Ours** | **0.816** | **0.032** | **0.915** | **0.038** | **0.866** | **0.035** |

*Table 5.* Generation quality and pose/expression fidelity on CelebA.

| Method | FID ↓ | FD ↑ | Yaw ↓ | Pitch ↓ | Roll ↓ | Expr ↑ |
|---|---|---|---|---|---|---|
| IVFG | 186.685 | **1.0** | 15.266 | 6.003 | 3.740 | 0.507 |
| DP2 | 94.305 | 0.989 | 5.488 | 4.065 | 3.653 | 0.594 |
| RiDDLE | 99.178 | 0.999 | 4.504 | 4.298 | 2.414 | 0.657 |
| KFAAR | 187.559 | 0.999 | 4.150 | 4.050 | 2.688 | 0.680 |
| G$^2$Face | 77.503 | 0.991 | 3.611 | 3.753 | 2.544 | 0.690 |
| FAS | 68.462 | 0.992 | 3.391 | 3.695 | 2.397 | 0.738 |
| **Ours** | **33.530** | **1.0** | **3.249** | **3.028** | **2.286** | **0.780** |

*Table 6.* Watermark forgery attack ASR (%), inversion-and-regeneration (I&R) and deepfake face-swap (FS).

| Method | I&R | FS |
|---|---|---|
| Tree-Ring | 33.4 | 91.8 |
| Gaussian Shading | 100.0 | 100.0 |
| WIND | 100.0 | 100.0 |
| PRC | 100.0 | 81.4 |
| **Ours (IGGW)** | **1.5** | **18.2** |

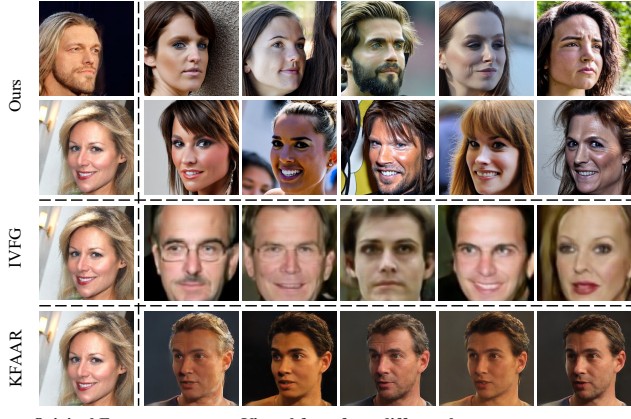

*Figure 3.* Diverse virtual face images generated from the same original face under different keys.

and PID recoverability, indicating that the generated VIDs preserve utility while limiting recoverable PID cues.

**Diversity.** To evaluate diversity, we measure the pairwise mismatch rate among virtual face images generated from the same original face image. Specifically, for each original face image, we generate four virtual face images using different control settings. The pairwise mismatch rate is computed by comparing all $\binom{4}{2}$ pairs of virtual faces and counting the fraction of pairs whose cosine similarity is below the EER threshold. We also report the mean cosine similarity (mCos), where lower mCos indicates higher diversity.

As shown in Table 4, our method achieves substantially higher diversity than prior baselines on both datasets. This advantage is also visually confirmed in Fig. 3, compared with two representative IVF methods, IVFG and KFAAR, our results exhibit noticeably higher face quality and richer cross-key appearance variations, producing more realistic and diverse virtual faces under different keys.

**Generation quality and pose/expression fidelity.** The generation quality of our method is shown in Fig. 2 and Fig. 3. It can be seen that our method produces photorealistic virtual faces while largely preserving the pose and facial expression of original faces.

We further conduct quantitative evaluation on CelebA. The Fréchet Inception Distance (FID) (Heusel et al., 2017) and

face detection rate (FD) using RetinaFace (Deng et al., 2020) are used to evaluate the basic quality of face images. To measure pose and expression preservation, we use the Face++ API to extract head-pose angles (Yaw, Pitch, Roll) and expression attributes (Expr) from both the input image $I$ and the generated image $\hat{I}$. For head pose, we compute the absolute angle differences, for expression, we normalize the attribute vectors and then measure their cosine similarity. The results are reported in Table 5. It can be seen that compared with all baselines, our method achieves better visual quality while yielding lower pose errors and higher expression similarity.

**Efficiency.** We evaluate the computational cost of different methods to assess practical deployability. On an RTX 4090 GPU, TIVDiff takes 1.37 s to generate a $512 \times 512$ virtual face image, and IGGW verification introduces only 0.68 s additional overhead. In comparison, the recent diffusion-based method FAS (Kung et al., 2025) requires 4.05 s for generation. These results show that TIVDiff achieves high-fidelity and trustworthy IVF generation while maintaining practical inference efficiency.

### 4.3. Watermark Evaluation

**Verification effectiveness.** To test the effectiveness of our verification system in real-world settings, we conduct verification experiments on clean images and common post-

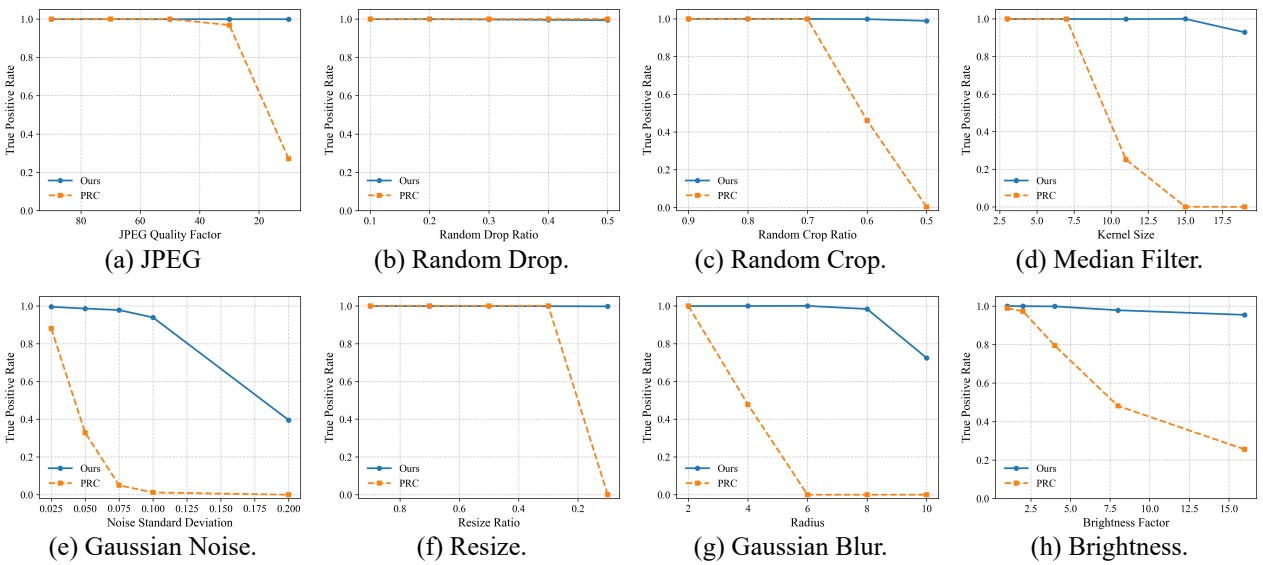

*Figure 4.* Verification robustness under common post-processing. TPR at a fixed FPR of $10^{-6}$ under different post-processing.

*Table 7.* VIP loss ablation. Each variant removes one VIP loss term at a time while keeping the remaining terms fixed, to quantify the contribution of each loss.

| Methods | AUC | EER | FID | PR | Div. |
|---|---|---|---|---|---|
| w/o $L_{\text{anony}}$ | 0.988 | 0.037 | 32.130 | 0.844 | 0.861 |
| w/o $L_{\text{div}}$ | 0.923 | 0.146 | 40.363 | 0.846 | 0.001 |
| w/o $L_{\text{intra}}$ | 0.941 | 0.122 | 29.359 | 0.966 | 0.879 |
| w/o $L_{\text{inter}}$ | 0.980 | 0.055 | 36.205 | 0.971 | 0.861 |
| w/o $L_{\text{dis}}$ | 0.963 | 0.092 | 73.503 | 0.980 | 0.880 |
| **Ours** | 0.988 | 0.036 | 33.530 | 0.979 | 0.881 |

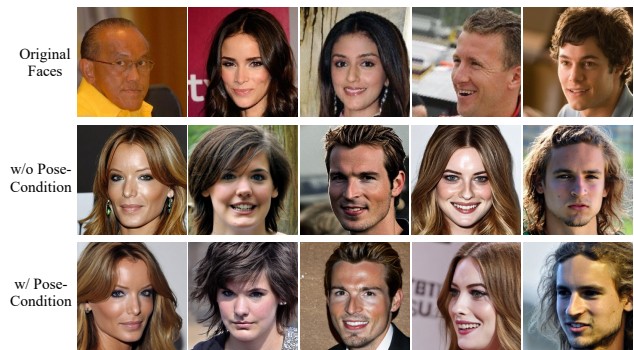

*Figure 5.* Examples of our virtual face images generated with and without the pose condition.

processing with different intensities. For clean setting and each post-processing intensity level, we generate 2,000 image pairs with and without IGGW, and report the true positive rate (TPR) at a fixed false positive rate (FPR) of $10^{-6}$. For clean images, the verification achieves perfect performance with TPR = 1.0. For post-processing images, we compare our IGGW with PRC (Gunn et al., 2025). The results are reported in Fig. 4. It can be seen that aggressive Gaussian noise and strong Gaussian blur gradually reduce TPR of both methods, and our method remains more robust than PRC against all post-processing.

**Security against watermark forgery attacks.** We consider realistic watermark forgery attacks where an adversary obtains a watermarked face image and transplants the same valid watermark to mass-produce unauthorized faces. The goal is to make these unauthorized faces accepted as legitimate, thereby undermining the trustworthiness of legitimate virtual faces. Concretely, such watermark forgery attacks can be instantiated in two practical ways, including (i) inversion-and-regeneration, where a stronger adversary with access to the private generator weights runs an

inversion-and-regeneration pipeline to reattach an initial noise watermark onto unauthorized face images, and (ii) deepfake face-swap, where an adversary performs a face-swap manipulation that replaces legitimate virtual faces with unauthorized faces while preserving the global watermark signal. Both instantiations can cause unauthorized faces to be falsely accepted.

We generate 2,000 image pairs for each watermark method (IGGW and other baselines), each pair consists of a legitimate sample and the corresponding forged sample. We report the attack success rate (ASR), defined as the rate of forged samples that are accepted as legitimate. Table 6 summarizes the ASR for both attacks, where lower values indicate stronger resistance. It can be seen that most baselines suffer from high ASR, with several methods completely failing. Compared to the best-performing baseline in each setting, our method achieves superior performance, reducing ASR by 31.9% for I&R and by 63.2% for FS, respectively.

## 4.4. Ablation Study

**VIP objective ablation.** We isolate the contribution of each VIP loss by removing one term at a time while keeping all others unchanged, including $w/o\ L_{\text{anony}}$, $w/o\ L_{\text{div}}$, $w/o\ L_{\text{intra}}$, $w/o\ L_{\text{inter}}$, and $w/o\ L_{\text{dis}}$. As shown in Table 7, removing $L_{\text{div}}$ nearly collapses diversity down to 0.001, while removing $L_{\text{anony}}$ weakens privacy protection. Removing $L_{\text{intra}}$ or $L_{\text{inter}}$ increases EER, highlighting the importance of consistency and separability for stable VID assignment. Moreover, dropping the distribution regularization term $L_{\text{dis}}$ sharply worsens FID, indicating that distribution alignment is crucial for realistic generation. Overall, the full objective achieves the best trade-off among privacy, identifiability, diversity, and image quality.

**Pose condition ablation in generation.** We further study the impact of the pose condition by comparing generation with and without pose condition. As visualized in Fig. 5, disabling pose condition leads to noticeable drift in head pose and expression relative to the input, while enabling the pose condition better preserves the input pose/expression cues, improving controllability and perceptual fidelity.

## 5. Conclusion

In this paper, we propose a Trustworthy Identifiable Virtual Face Diffusion (TIVDiff) framework for trustworthy and identifiable virtual face (IVF) generation. Our TIVDiff first maps the user's physical identity (PID) to diverse virtual identities (VIDs) via a Virtual Identity Projection (VIP). Then, a Pose-Aware Identity-Conditioned Diffusion (PoseID-Diff) module is guided by a selected VID and the 3D facial geometry of the original face to generate a high-quality IVF, which preserves the original pose and expression for challenge-response requirements. For trustworthiness, we propose an Identity-Guarded Generative Watermarking (IGGW) scheme, which enables the system to verify the legitimacy of IVF by a reversible mapping between the VID and the diffusion initial noise. Extensive experiments show that the proposed TIVDiff enables trustworthy and controllable IVF generation, with quality and identifiability superior to the SOTA IVF schemes. Under common post-processing and watermark forgery attacks, the IGGW is more robust than SOTA watermark schemes.

## Acknowledgment

This work was supported by Ant Group through CCF-Ant Research Fund.

## Impact Statement

This paper presents work whose goal is to advance the field of Machine Learning. There are many potential societal consequences of our work, none which we feel must be specifically highlighted here.

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
