# OpenReview forum: "Towards Trustworthy and Identifiable Virtual Face Generation"
_ICML.cc/2026/Conference — ICML 2026 regular_

### Official Review · Reviewer_dvdE · 2026-03-11

**Soundness:** 2
**Presentation:** 2
**Significance:** 2
**Originality:** 2
**Overall Recommendation:** 4
**Confidence:** 4

**Summary:**

This paper proposes TIVDiff, a diffusion-based framework for Identifiable virtual face (IVF) trustworthiness generation. To enable trustworthiness of the IVF, this paper proposes IGGW to bind the diffusion initial noise with VID through a reversible mapping. Experiment results demonstrate the proposed TIVDiff outperforms GAN-based IVF generation framework in terms of image quality, identifiability and trustworthiness.

**Compliance With Llm Reviewing Policy:**

Affirmed.

**Final Justification:**

Thank you for the author's response, which resolved my issue. I am inclined to accept the paper weakly.

**Key Questions For Authors:**

Please refer to the weaknesses.

**Limitations:**

No. Please present the failure cases and discuss why these cases are failure.

**Strengths And Weaknesses:**

Strengths:
This paper tackles how to make privacy-preserving synthetic identities trustworthy, which is worth to research. The proposed IGGW provides reliable verification than using a static pattern. The experiments covering many tasks, such as virtual face indentifiability, anonymization, diversity, generation quality, and watermark.

Weaknesses:
1. Lacking comparing with diffusion-based baselines: In Table 1 to Table 5, the paper mainly compares their proposed method with several previous methods which based on GAN. However, these baselines are proposed in 2022, 2023, 2024, and one baseline named FAS in 2025. Besides, most of them are based on GAN. Thus, it difficult to determine from which aspect the performance gain is brought by IGGW or the diffusion architecture. I think it needs to compare with recent diffusion-based methods, such as [1] and [2].

2. Lacking experimental or theoretical analysis about the claim from L203 to L208: The paper claims “ In IGGW, we bind the diffusion initial noise with γ through a newly designed Reversible Distribution-Matched Mapping (RDMM), ensuring that the generated IVF based on this initial noise carries an intrinsic and verifiable legitimacy marker”. However, the paper did not provide any relative experiments, analysis, or citations to explanation why it can ensure “carries an intrinsic and verifiable legitimacy marker”.

3. Lacking experiments analysis: In table 1, the proposed method generates virtual face image even outperforms the original face images on identifiability level. This experimental result is somewhat counterintuitive, but the paper did not provide deeper analysis why this situation happened.

[1]. VIGFace: Virtual Identity Generation for Privacy-Free Face Recognition dataset (ICCV 2025)
[2]. Diffusion-based Adversarial Identity Manipulation for Facial Privacy Protection (ACM MM 2025)

---

> ### Author Rebuttal · Authors · 2026-03-31
>
> ## Q1. On the lack of diffusion-based baselines.
> Thank you for this important comment. We conducted a careful review of the recent literature including **DiffAIM** (ACM MM 2025) and **VIGFace** (ICCV 2025). The current official repository of VIGFace does not yet provide a fully reproducible implementation. We are unable to include a strict apples-to-apples reproduction of VIGFace within the rebuttal period.
>
> Regarding **DiffAIM**, we include it in our comparison under the subset of metrics and evaluation protocols that can be fairly aligned with our IVF setting. The comparison results with our method are reported below, where TIVDiff outperforms DiffAIM on most metrics.
>
> | Method  | ArcFace EER ↓ | ArcFace AUC ↑ | AdaFace EER ↓ | AdaFace AUC ↑ |
> |---------|----------------|---------------|----------------|---------------|
> | DiffAIM |0.088|0.970|0.106|0.955|
> | Ours    |0.036|0.988|0.045|0.986|
>
> | Method  | ArcFace PR ↑ | ArcFace mCos ↓ | AdaFace PR ↑ | AdaFace mCos ↓ |
> |---------|---------------|----------------|--------------|----------------|
> | DiffAIM |0.002|0.439|0.003|0.415|
> | Ours    |0.979|-0.024|0.981|0.002|
>
> | Method  | ArcFace Div. ↑ | ArcFace mCos ↓ | AdaFace Div. ↑ | AdaFace mCos ↓ |
> |---------|-----------------|----------------|----------------|----------------|
> | DiffAIM |0.001|0.559|0.001|0.520|
> | Ours    |0.881|0.025|0.928|0.037|
>
> | Method  | FID ↓ | FD ↑ | Yaw ↓ | Pitch ↓ | Roll ↓ | Expr ↑ |
> |---------|-------|------|-------|---------|--------|--------|
> | DiffAIM |49.743|1.0|3.488|3.308|2.251|0.790|
> | Ours    |33.530|1.0|3.249|3.028|2.286|0.780|
>
> We speculate that DiffAIM’s low privacy and diversity. One possible reason is that its adversarial anonymization mechanism, whose effect may not transfer well to the ArcFace and AdaFace used in our experiments, thereby leaving relatively high identity similarity.
>
>
>
> ## Q2. On the claim that RDMM makes the generated IVF carry an intrinsic and verifiable legitimacy marker.
>
> We thank the reviewer for this important comment. To support the claim that RDMM provides an intrinsic and verifiable legitimacy marker, we add both a **theoretical intuition** and a **distribution-matching analysis**.
>
> ### Theoretical intuition
>
> Our verification checks the consistency between the **noise-side recovered direction** and the **image-side identity embedding**. Under a **mismatch** case, these two directions can be approximated as independent random unit vectors in $\mathbb{R}^{512}$. Let the verification score be
> $$
> s = \langle u, v \rangle,
> $$
> where $u, v \in \mathbb{S}^{512}$ are two independent unit vectors.
>
> In high dimension, the cosine similarity between two independent random directions concentrates around zero, with
> $$
> \mathbb{E}[s] = 0, \qquad \mathrm{Var}(s) = \frac{1}{512}.
> $$
> Thus,
> $$
> s \approx \mathcal{N}\left(0,\frac{1}{512}\right).
> $$
>
> Therefore, for a fixed threshold $\tau > 0$, the false-accept probability under mismatch is
> $$
> \Pr(s > \tau) \approx 1 - \Phi(\tau \sqrt{512}),
> $$
> which is very small. This explains why the legitimacy marker is verifiable: matched samples yield high similarity, while mismatched samples remain concentrated near zero.
>
>
> ### Distribution-matching analysis
>
> We further test whether RDMM preserves Gaussian-compatible statistics required by diffusion initialization. Specifically, we fix a base vector in $\mathbb{R}^{512}$, generate $10{,}000$ RDMM outputs under different keys, normalize them to unit length, and compute cosine similarities over $2{,}000{,}000$ randomly sampled pairs.
>
> If RDMM preserves the desired spherical-direction behavior, the empirical cosine-similarity distribution should match
> $$
> \mathcal{N}\left(0,\frac{1}{512}\right).
> $$
>
> The empirical results closely follow this theoretical distribution in terms of mean, standard deviation, and representative quantiles:
>
> | Quantile | Theoretical $\mathcal{N}(0,1/512)$ | Empirical |
> |---|---:|---:|
> | 1% | -0.1028 |-0.1028|
> | 5% | -0.0726 |-0.0728|
> | 50% | 0.0000 |-1.355e-5|
> | 95% | 0.0726 |0.0726|
> | 99% | 0.1028 |0.1027|
>
> | Statistic | Theoretical | Empirical |
> |---|---:|---:|
> | Mean | 0 |-4.534e-5|
> | Std | $1/\sqrt{512} \approx 0.0442$ | 0.0442 |
>
> These results indicate that RDMM remains sufficiently close to Gaussian initialization for diffusion generation, while still preserving reversible identity-coupled directional information for legitimacy verification.
>
> ## Q3. Why does Table 1 show identifiability exceeding the original faces?
> Please refer to **Reviewer K9Y8, Q2**.

---

> > ### Author Rebuttal · Reviewer_dvdE · 2026-04-08
> >
> > Thank you for the author's response, which resolved my issue. I am inclined to accept the paper weakly.

---

### Official Review · Reviewer_8u5n · 2026-03-12

**Soundness:** 3
**Presentation:** 3
**Significance:** 3
**Originality:** 3
**Overall Recommendation:** 4
**Confidence:** 4

**Summary:**

The authors argue that **privacy protection and trustworthiness have been largely overlooked in current Identifiable Virtual Face (IVF) generation**, which may introduce potential risks associated with deepfake attacks. To address this issue, they propose the **TIVDiff (Trustworthy Identifiable Virtual Face Diffusion) framework**, which aims to generate **verifiable and trustworthy virtual faces**. Specifically, the framework first introduces a **Virtual Identity Projection (VIP)** module that maps a real face's identity representation into a **Virtual Identity (VID) space**, ensuring the virtual face is visually different from the real one while preserving identity consistency. It then employs a **Pose-Aware Identity-Conditioned Diffusion (PoseID-Diff)** module that injects both virtual identity information and 3D facial geometry cues into the diffusion model to generate high-quality faces while maintaining the original pose and expression. Finally, an **Identity-Guarded Generative Watermarking (IGGW)** mechanism embeds the virtual identity into the initial noise of the diffusion process, enabling reversible watermarking for verifying the legitimacy of generated virtual faces without revealing the real identity. Overall, the main contribution of this work lies in **integrating watermark-based verification with virtual face generation**, which can be viewed as applying existing techniques to a new application scenario. Therefore, my rating is **Weak Accept**.

**Compliance With Llm Reviewing Policy:**

Affirmed.

**Final Justification:**

I have no other questions and keep my rating (*weak accept*).

**Key Questions For Authors:**

See the content in **Weakness** and **Minor Weakness**

**Limitations:**

yes

**Strengths And Weaknesses:**

**Strength**
1. The proposed method, **TIVDiff**, improves the privacy and trustworthy issues in existing **Identifiable Virtual Face (IVF)** generation methods. By integrating watermark embedding and verification into the IVF generation process, the method ensures that malicious deepfake attacks cannot pass the verification system.
2. The authors provide extensive experiments demonstrating that the proposed method can achieve high generation quality, while the generated IVFs simultaneously satisfy both **verifiability** and **identity consistency** requirements.
3. The paper is clearly written and easy to follow.

**Weakness**
1. The paper lacks security evaluations under potential attack scenarios. In general, a secure system should evaluate whether its privacy or security properties can still be maintained under different types of attacks (for example, adaptive adversarial attacks are commonly introduced when evaluating adversarial defense methods). For instance, if an attacker obtains the noise map generated from the VID, is it possible to recover or simulate the embedded watermark through adversarial attacks or other techniques? Can authors further provide some experiments or discussion?

**Minor Weakness**
1. There appears to be a typo in **Equation (6)**. Should the $r$ in the equation actually be $q$?
2. The paper lacks references to existing **face generation methods**, such as **Arc2Face [1]**, which have been widely used in diffusion-based face generation conditioned on face identity.
3. The authors may consider further explaining why the generated images are required to preserve attributes from the original image that are unrelated to identity, such as **pose** and **facial expressions**.

[1]. Papantoniou F P, Lattas A, Moschoglou S, et al. Arc2face: A foundation model for id-consistent human faces[C]//European Conference on Computer Vision, 2024: 241-261.

---

> ### Author Rebuttal · Authors · 2026-03-31
>
> ## Q1. Security evaluations under potential attack scenarios.
> We thank the reviewer for this helpful suggestion. In the current submission, we evaluate two realistic watermark forgery attacks, inversion-and-regeneration and face-swap, and show that IGGW is substantially more robust than prior methods under these practical attack scenarios. To further assess resistance to adaptive attackers, we additionally implement two adaptive attack settings tailored to IGGW, which checks the consistency between the recovered noise-side watermark evidence and the image-side identity embedding.
>
> Specifically, we consider:
>
> 1. **Adaptive joint optimization**, which directly optimizes a joint verifier-oriented objective combining the noise-side consistency score and the image-side identity consistency score.
>
> 2. **Adaptive surrogate verification**, which first learns a lightweight surrogate model to approximate the verifier score and then optimizes the forged image against this surrogate.
>
> In our implementation, the adaptive joint attacker uses a score-based NES optimization procedure over a bounded image-space perturbation set, while the adaptive surrogate attacker alternates between querying the true verifier, updating the surrogate, and performing gradient-based optimization against the learned surrogate.
>
> For the adaptive joint attacker, we optimize
>
> $$
> \mathcal{J}(I) = \alpha \, s_{\text{noise}}(I) + \beta \, s_{\text{img}}(I),
> $$
>
> where $s_{\text{noise}}(I)$ denotes the consistency between the target VID and the VID recovered from the inverted initial noise, and $s_{\text{img}}(I)$ denotes the consistency between the target VID and the image-side face embedding extracted from the attacked image. We set $\alpha = \beta = 0.5$, and use NES-based score optimization under an $\ell_{\infty}$ budget of $\epsilon = 8/255$, with step size $2/255$, smoothing parameter $1/255$, 6 optimization steps, and 8 NES samples per step.
>
> For the adaptive surrogate attacker, we use the same joint objective, but approximate it with a learned surrogate verifier. Concretely, the attacker first collects queried image-score pairs, trains a lightweight CNN surrogate to regress the verifier score, and then performs iterative gradient-based optimization on the image using the surrogate, while periodically refreshing the training set with additional true verifier queries. In our implementation, we use the same $\ell_{\infty}$ budget $\epsilon = 8/255$, step size $1/255$, 6 outer iterations, 5 inner optimization steps, 12 bootstrap queries, and 6 local queries per outer iteration.
>
> We evaluate these adaptive attacks using the same security metric as in the main paper, namely attack success rate (ASR), defined as the percentage of forged samples that are falsely accepted as legitimate by the verifier. Under our current adaptive attack implementations, we do not observe successful forgeries for either attacker, and the ASR is 0.0% for adaptive joint optimization and 0.0% for adaptive surrogate verification. A likely explanation for the zero ASR is that IGGW verifies watermark evidence through diffusion inversion, the attacker must not only perturb the image, but also induce a precise change in the initial noise recovered from that image. This makes the attack objective highly indirect and nonlinear in image space, so small bounded perturbations are insufficient to reliably produce successful forgeries under our tested budgets.
>
> ## Q2. The questions about Eq. (6).
> Thank you for pointing out this potential ambiguity. The purpose of Eq. (6) is to enforce inter-identity separability under a fixed key, i.e., different physical identities (PIDs) should still produce distinct virtual identities (VIDs) even when the same key is used. In our notation, $p$ and $q$ denote samples from the same PID, while $p$ and $r$ denote samples from different PIDs. We realize that this was not explained clearly enough around the equation, which may have caused the confusion. We will revise the text around Eq. (6) to define these sample relationships explicitly and improve clarity.
>
> ## Q3. Clarifications on the Arc2Face reference.
> Thank you for this suggestion. Arc2Face is used to initialize PoseID-Diff in our implementation and is cited in Section 4.1 (Experimental Setup, Implementation Details).
>
> ## Q4. Why preserve pose / expression?
> Modern face recognition systems, especially under challenge-response protocols, often require users to perform natural variations in head pose and facial expression. Our goal is therefore not to generate a static identifiable virtual face (IVF), but an IVF that can still reflect user-driven pose and expression changes while maintaining identity consistency. We will strengthen the explanation in both the Introduction and Method sections.

---

> > ### Author Rebuttal · Reviewer_8u5n · 2026-04-03
> >
> > Most of my concern has been addressed. I keep my rating.

---

### Official Review · Reviewer_K9Y8 · 2026-03-13

**Soundness:** 3
**Presentation:** 3
**Significance:** 2
**Originality:** 2
**Overall Recommendation:** 4
**Confidence:** 5

**Summary:**

This paper targets identifiable virtual face generation for privacy protection, where a user’s original face is transformed into a visually different but still identity-usable virtual face. The authors propose TIVDiff, a diffusion-based framework with three main components: (1) Virtual Identity Projection that maps a PID embedding to multiple selectable VID embeddings under a user-specific key while enforcing anonymization, diversity, intra-identity consistency, and inter-identity separability; (2) Pose-Aware Identity-Conditioned Diffusion built on Stable Diffusion that injects VID into the text prompt and uses 3D facial geometry via a ControlNet branch to preserve pose/expression; and (3)Identity-Guarded Generative Watermarking  that binds the diffusion initial noise to the VID through a reversible distribution-matched mapping, enabling legitimacy verification by inverting a queried image to recover its initial noise and checking consistency between the noise-implied VID and image-side VID. Experiments on CelebA and FFHQ show improvements over IVF/anonymization baselines in image quality, identifiability, and verification robustness, including resistance to post-processing and watermark forgery attacks.

**Compliance With Llm Reviewing Policy:**

Affirmed.

**Final Justification:**

Most of my concern has been addressed. I have increased my score to WA.

**Key Questions For Authors:**

1. Have you evaluated stronger privacy attacks, such as cross-key linkage, PID inference from VID/IVF embeddings, or training a classifier to link IVF to PID? A positive result here would materially increase confidence in privacy claims.
2. Why identifiability exceeds Original: Table 1 suggests improved identifiability over original faces. Can you explain the mechanism (e.g., denoising/normalization effects, distribution shift) and validate it across additional recognizers and datasets?
3.Your forgery attacks include inversion-and-regeneration and face-swap. Have you considered adaptive attackers that explicitly optimize to match the noise-side and image-side VID consistency (e.g., joint optimization or learned surrogate verification)? How resilient is IGGW under such adaptive attacks?

**Limitations:**

yes

**Strengths And Weaknesses:**

Strengths
1. Well-motivated decomposition. VIP (identity-space design), PoseID-Diff (generation with pose/expression control), and IGGW/RDMM (verification) form a clean end-to-end pipeline, aligned with the stated goals: privacy, utility, controllability, and legitimacy.

Weaknesses
1. Privacy analysis may be incomplete**: While VIP pushes VID away from PID and controls diversity/consistency, it remains unclear how well the method resists advanced linkage attacks (e.g., cross-key linkage, membership inference, or reconstruction of PID from VID/IVF), beyond cosine metrics.
2. The Table 1 suggests the method surpasses original faces in identifiability, which is surprising. This may be due to generation bias, data split, or recognizer behavior; it deserves additional analysis to avoid over-claiming.
3. Experiments are on CelebA and FFHQ; broader settings (different demographics, sensors, lighting, occlusions, video-based challenge-response, or in-the-wild capture) would better validate method effectiveness.

---

> ### Author Rebuttal · Authors · 2026-03-31
>
> ## Q1. Stronger privacy attacks evaluation.
> In the paper, we evaluate the privacy-preserving capability of the proposed method by measuring the mismatch rate between the original faces and their corresponding virtual faces. To further evaluate our method, we add stronger privacy analyses, including **cross-key linkage**, **PID inference**, **membership inference**, and **PID reconstruction** attacks.
>
> Specifically:
>
> - **Cross-key linkage** evaluates whether two anonymized features generated under different keys can still be linked to the same identity.
>   We report **AUC** and **Rank-1 Linkage**. Lower values indicate that anonymized features are harder to associate across keys.
>
> - **PID inference** evaluates whether the original identity can still be inferred from the anonymized feature.
>   We report **Top-1 accuracy** for classification-based attacks and **Rank-1** for retrieval-based attacks. Lower values indicate better identity protection.
>
> - **Membership inference** evaluates whether an attacker can distinguish training members from non-members from model outputs.
>   We report **AUCs** for four standard confidence-based attacks. An AUC closer to chance indicates weaker membership leakage.
>
> - **PID reconstruction** evaluates a stronger adversary that attempts to reconstruct PID-related features from the anonymized representation.
>   We report **cosine similarity** between reconstructed and true PID features, and **PID Rank-1** based on the reconstructed features. Lower values indicate that recovering original identity information is more difficult.
>
> We conduct all attacks on the **out-of-domain FFHQ** dataset and use **AdaFace** as the attacker’s identity feature encoder. Since **AdaFace is not involved in our method**, these settings provide a stronger robustness check for the privacy evaluation.
>
> The results are reported in the following tables. Our method consistently exhibits stronger privacy protection under  evaluated attacks. The significantly lower attack performance demonstrates that TIVDiff more effectively limits recoverable identity cues and is therefore more robust against privacy leakage.
>
> ### Cross-key linkage results
> | Method | AUC ↓ | Rank-1 ↓ |
> |---|---:|---:|
> | IVFG | 0.6052 | 0.0323 |
> | RiDDLE | 0.8245 | 0.7455 |
> | KFAAR | 0.9806 | 0.8512 |
> | G2Face | 0.8120 | 0.3845 |
> | Ours | 0.4769 | 0.0027 |
>
> ### PID Recovery Attack Results
>
> #### Classification-based Attacks
>
> | Method | Top-1 Acc ↓ |
> |---|---:|
> | IVFG | 0.0110 |
> | RiDDLE | 0.6170 |
> | KFAAR | 0.7930 |
> | G2Face | 0.4600 |
> | Ours | 0.0010 |
>
> #### Retrieval-based Attacks
>
> | Method | Rank-1 ↓ |
> |---|---:|
> | IVFG | 0.0018 |
> | RiDDLE | 0.0245 |
> | KFAAR | 0.0010 |
> | G2Face | 0.0163 |
> | Ours | 0.0003 |
>
> ### Membership inference
>
> | Method | AUC(max conf) ↓ | AUC(true conf) ↓ | AUC(-entropy) ↓ | AUC(-CE loss) ↓ |
> |---|---:|---:|---:|---:|
> | IVFG | 0.7396 | 0.9281 | 0.7188 | 0.9281 |
> | RiDDLE | 0.9319 | 0.9212 | 0.9310 | 0.9212 |
> | KFAAR | 0.9226 | 0.9161 | 0.9226 | 0.9161 |
> | G2Face | 0.9376 | 0.9224 | 0.9351 | 0.9224 |
> | Ours | 0.5376 | 0.7681 | 0.5338 | 0.7681 |
>
> ### PID reconstruction
>
> | Method | CosSim ↓ | PID Rank-1 ↓ |
> |---|---:|---:|
> | IVFG | 0.1256 | 0.0270 |
> | RiDDLE | 0.4055 | 0.8620 |
> | KFAAR | 0.3506 | 0.7600 |
> | G2Face | 0.4464 | 0.9620 |
> | Ours | 0.1296 | 0.0220 |
>
> ## Q2. Clarifications on the claim. Why does Table 1 show identifiability exceeding the original faces?
> The main reason is that we assign each pid a unique key to generate the vid. the key serves as a user-specific factor that encourages VIDs from the same PID to be more compact in the identity space. In contrast, the original faces may have more intra-class variation due to factors such as pose, expression, and lighting, which can lead to lower identifiability. By using the same key for different face samples of the same PID, we effectively reduce this intra-class variation in the virtual identity space, leading to higher identifiability for the generated IVFs compared to the original faces.
>
> To further verify that this phenomenon, we additionally evaluate identifiability using **FaceNet** model and LFW dataset. The results are shown below.
> | Dataset | Recognizer | EER ↓ | AUC ↑ |
> |---------|------------|-------|-------|
> | Original Faces (CelebA) | FaceNet | 0.047 | 0.975 |
> | Generated IVFs (CelebA) | FaceNet | 0.037 | 0.979 |
> | Original Faces (LFW)    | ArcFace | 0.041 | 0.979 |
> | Generated IVFs (LFW)    | ArcFace | 0.039 | 0.984 |
> | Original Faces (LFW)    | AdaFace | 0.049 | 0.977 |
> | Generated IVFs (LFW)    | AdaFace | 0.043 | 0.983 |
> ## Q3. Generalization beyond CelebA and FFHQ.
> Please refer to **Reviewer rWDK, Q3**.
>
> ## Q4. Adaptive attacker evaluation.
> To further assess resistance to adaptive attackers, we additionally implement two adaptive attack settings, including **Adaptive joint optimization** and **Adaptive surrogate verification**. Please refer to **Q1 of Reviewer 8u5n**.

---

> > ### Author Rebuttal · Reviewer_K9Y8 · 2026-04-06
> >
> > Most of my concern has been addressed.  I have increased my score to WA.

---

### Official Review · Reviewer_rWDK · 2026-03-18

**Soundness:** 3
**Presentation:** 3
**Significance:** 3
**Originality:** 2
**Overall Recommendation:** 5
**Confidence:** 3

**Summary:**

This research focuses on privacy-preserving face recognition using generative models and proposes a framework called TIVDiff (Trustworthy Identifiable Virtual Face Diffusion) for generating identifiable virtual faces (IVFs). The goal is to transform an original face into a virtual one that protects privacy while preserving identity information for recognition systems. The framework includes three modules: Virtual Identity Projection (VIP) for mapping physical identity to virtual identity, Pose-Aware Identity-Conditioned Diffusion (PoseID-Diff) for generating high-quality virtual faces, and Identity-Guarded Generative Watermarking (IGGW) for embedding a reversible watermark to verify authenticity. Experiments on the CelebA and FFHQ datasets demonstrate improvements in identifiability, privacy protection, diversity, and watermark robustness compared to previous methods.

**Compliance With Llm Reviewing Policy:**

Affirmed.

**Key Questions For Authors:**

1. The paperis expected to be benefitted with stronger theoretical grounding, as it lacks formal privacy guarantees, in-depth identity leakage analysis, and evaluation of runtime/latency, which are important for practical deployment.
2. The approach may incur high computational costs, and the reliance on datasets such as CelebA and FFHQ raises concerns about how well the results generalize to real-world, unconstrained scenarios.
3. The evaluation of watermark robustness against attacks is somewhat limited, and the discussion on ethical and societal implications is relatively brief and could be expanded.

**Limitations:**

1. The presented method lacks formal privacy guarantees and thorough identity leakage analysis, limiting confidence in its robustness against strong adversaries.
2. The approach is likely computationally expensive, with no reported runtime or latency evaluation, raising concerns about real-world deployment and scalability.
3. The evaluation presnted is limited to curated datasets (CelebA, FFHQ) and constrained attack settings, leaving uncertainty about generalization, real-world robustness, and watermark security.

**Strengths And Weaknesses:**

1. The paper addresses an important problem of privacy-preserving face recognition with a well-structured framework (VIP, PoseID-Diff, IGGW) where it uses diffusion models to achieve higher-quality virtual faces and shows strong performance improvements over prior IVF methods.
2. It includes experiments evaluating identifiability, anonymization, diversity, watermark security, and resistance to attacks such as inversion–regeneration and face swapping.
3. The novelty in the paper is somewhat incremental, mainly integrating existing techniques like diffusion models, ControlNet, and watermarking.
4. However, it lacks formal privacy guarantees, detailed identity-leakage analysis, and runtime/latency evaluation for real-world systems.
5. This approach may have high computational cost, and experiments on CelebA and FFHQ may not fully represent real-world scenarios.
6. Watermark attack evaluation is limited, and the discussion of ethical and societal impacts is brief.

---

> ### Author Rebuttal · Authors · 2026-03-31
>
> ## Q1. Clarifications on the novelty and contribution.
> Existing GAN-based methods for Identifiable Virtual Face (IVF) generation often suffer from limited fidelity and weak controllability. They also lack the ability to verify the legitimacy of a generated virtual face, a key capability for practical deployment. To address this, we propose a Trustworthy IVF Diffusion (TIVDiff) framework that extends prior IVF research beyond the privacy-utility setting by enabling legitimacy verification without recovering the original physical identity (PID). TIVDiff comprises three key components: Virtual Identity Projection (VIP), Pose-Aware Identity-Conditioned Diffusion (PoseID-Diff), and Identity-Guarded Generative Watermarking (IGGW). VIP maps each PID to a diverse and user-selectable virtual identity (VID), with a key-controlled mechanism and multiple objectives that help ensure stable VID assignment. PoseID-Diff is introduced for controllable and high-fidelity IVF generation. In addition, unlike existing watermarking methods, IGGW is specifically developed for trustworthy IVF generation. It binds the VID to the initial diffusion noise through a novel Reversible Distribution-Matched Mapping (RDMM), so that the generated IVF carries verifiable watermark evidence.
>
> ## Q2. Privacy guarantees, identity leakage analysis and evaluation of runtime.
> Thanks for this valuable comment. Our VIP generates a key-conditioned VID and explicitly enforces separation from the PID through an anonymization objective, which is intended to reduce direct identity leakage and cross-key linkability. To further support this, we provide stronger privacy analyses, including cross-key linkage, PID inference, membership inference, and PID reconstruction. Please refer to **Q1 of Reviewer K9Y8** for details.
>
> We compare the computational cost of TIVDiff with two diffusion-based methods, as summarized below. On an RTX 4090 GPU, generating a 512 × 512 IVF with TIVDiff requires **1.37 s**, substantially lower than FAS and DiffAIM. Verifying IGGW adds only **0.68 s** of overhead.
>
> | Method | Batch inference | Generation time (s) | Verification time (s) | Peak GPU memory (GB) |
> |---|---|---:|---:|---:|
> | Ours |Yes|1.37|0.68|21.03|
> | FAS(2025) |Yes|4.05|-|22.96|
> | DiffAIM(2025) |No|148.07|-|21.12|
>
> ## Q3. Generalization in real-world scenarios beyond CelebA and FFHQ.
> Following prior studies, we conduct an out-of-domain evaluation on FFHQ in our paper. TIVDiff achieves state-of-the-art performance. To further evaluate generalization in more realistic scenarios, we conduct an evaluation on **LFW (Labeled Faces in the Wild)**, which contains more than 13,000 web-collected face images with substantial variation in pose, illumination, expression, background, and age. The same metrics used in the main paper, including identifiability, privacy, diversity, and visual quality are reported below. Our method achieves results on LFW comparable to the in-domain setting (i.e., CelebA), further validating its generalization ability.
>
> ### Identifiability
> | Dataset | ArcFace EER ↓ | ArcFace AUC ↑ | AdaFace EER ↓ | AdaFace AUC ↑ |
> |---------|----------------|---------------|----------------|---------------|
> | CelebA  |0.036|0.988|0.045|0.986|
> | LFW     |0.039|0.984|0.043|0.983|
> ### Privacy
> | Dataset | ArcFace PR ↑ | ArcFace mCos ↓ | AdaFace PR ↑ | AdaFace mCos ↓ |
> |---------|---------------|----------------|--------------|----------------|
> | CelebA  |0.979|-0.024|0.981|0.002|
> | LFW     |0.982|-0.022|0.986|0.012|
> ### Diversity
> | Dataset | ArcFace Div. ↑ | ArcFace mCos ↓ | AdaFace Div. ↑ | AdaFace mCos ↓ |
> |---------|-----------------|----------------|----------------|----------------|
> | CelebA  |0.881|0.025|0.928|0.037|
> | LFW     |0.867|0.025|0.925|0.034|
> ### Visual quality
> | Dataset | FID ↓ | FD ↑ | Yaw ↓ | Pitch ↓ | Roll ↓ | Expr ↑ |
> |---------|-------|------|-------|---------|--------|--------|
> | CelebA  |33.530|1.0|3.249|3.028|2.286|0.780|
> | LFW     |35.104|1.0|3.615|3.460|2.319|0.744|
>
> ## Q4. Watermark robustness under stronger attacks.
> Thanks for this helpful suggestion. To further demonstrate robustness of our watermark, we add stronger adaptive attack evaluations including **Adaptive joint optimization** and **Adaptive surrogate verification**. Please refer to **Q1 of Reviewer 8u5n** for details.
>
> ## Q5. Ethical and societal implications.
> Our method is designed for privacy-preserving and trustworthy identity virtualization, which may benefit applications where identity utility is needed without directly exposing real facial appearance. We acknowledge that such technology may raise broader societal concerns; for example, IVF systems could be misused in identity-related applications. We will expand the revised paper to discuss these potential risks and appropriate deployment safeguards.

---

### Decision · Program_Chairs · 2026-04-30

**Decision:**

Accept (regular)

**Comment:**

After the rebuttal, all of the reviewers gave the positive scores (4/4/5/4). Although Reviewer K9Y8 did not revise the score in the system, he/she replied that most of the concerns are addressed and will increase the score to Weak Accept.

After carefully reading the submission, the reviews, and the discussion, the reconmendation is: Accept.